# Via: Unified Spatiotemporal VIdeo Adaptation for Global and Local Video Editing

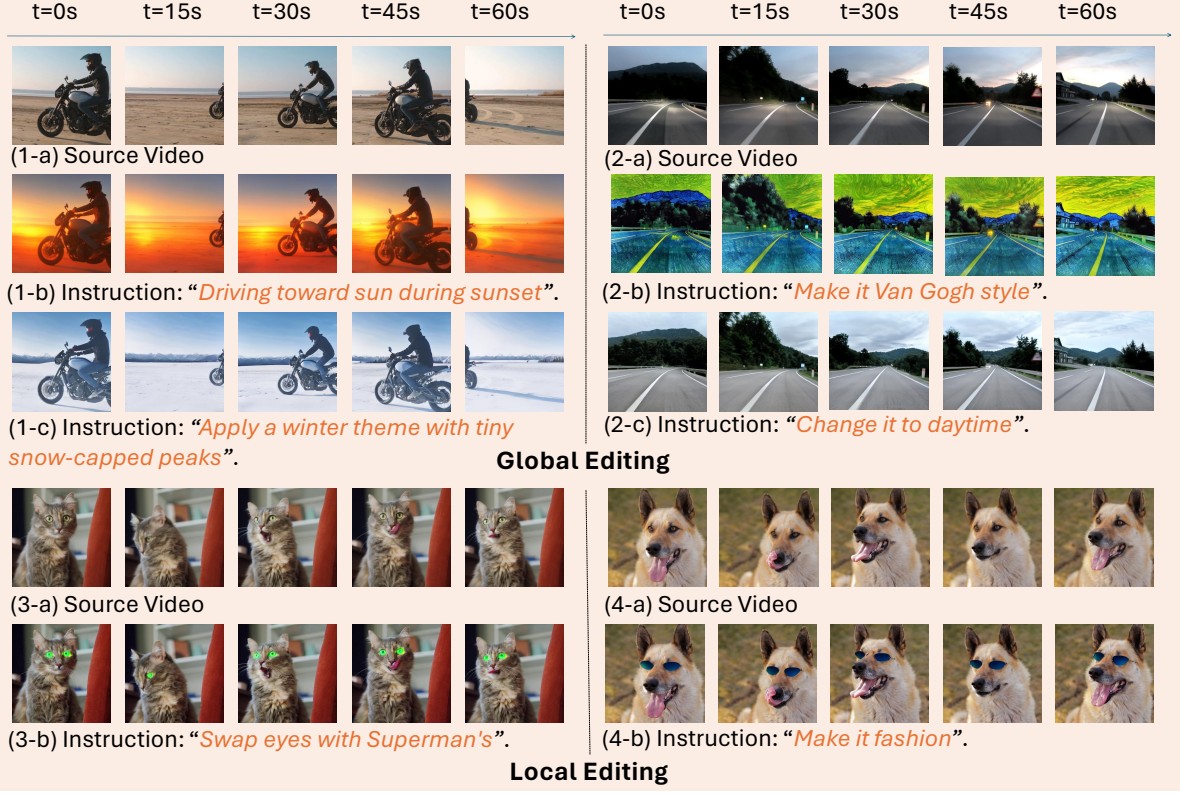

Figure 1: **Video editing results by Via.** VIA excels in *precise* and *consistent* editing across diverse video tasks. Top: consistent results over long videos with a duration of 1 minute, which is challenging in current literature. Bottom: consistent results for precise local editing.

## Abstract

Video editing serves as a fundamental pillar of digital media, spanning applications in entertainment, education, and professional communication. However, previous methods often overlook the necessity of comprehensively understanding both global and local contexts, leading to inaccurate and inconsistent edits in the spatiotemporal dimension, especially for long videos. In this paper, we introduce VIA, a unified spatiotemporal VIdeo Adaptation framework for global and local video editing, pushing the limits of consistently editing minute-long videos. First, to ensure local consistency within individual frames, we designed *test-time editing adaptation* to adapt a pre-trained image editing model for improving consistency between potential editing directions and the text instruction, and adapt masked latent variables for precise local control. Furthermore, to maintain global consistency over the video sequence, we introduce *spatiotemporal adaptation* that recursively *gather* con-

sistent attention variables in key frames and strategically applies them across the whole sequence to realize the editing effects. Extensive experiments demonstrate that, compared to baseline methods, our VIA approach produces edits that are more faithful to the source videos, more coherent in the spatiotemporal context, and more precise in local control. More importantly, we show that VIA can achieve consistent long video editing in minutes, unlocking the potential for advanced video editing tasks over long video sequences.

# 1 Introduction

With the exponential growth of digital content creation, video editing has become essential across various domains, including filmmaking (Frierson, 2018; Dancyger, 2018), advertising (Mei et al., 2007; Kholisoh et al., 2021), education (Calandra et al., 2008; 2009), and social media (Jackson, 2016; Schmitz et al., 2006). This task presents significant challenges, such as preserving the integrity of the original video, accurately following user instructions, and ensuring consistent editing quality across both time and space. These challenges are particularly pronounced in longer videos, where maintaining long-range spatiotemporal consistency is critical.

A substantial body of research has explored video editing models. One approach uses video models to process the source video as a whole (Ku et al., 2024; Liu et al., 2023b). However, due to limitations in model capacity and hardware, these methods are typically effective only for short videos (fewer than 200 frames). To overcome these limitations, various methods have been proposed (Xing et al., 2023; Wu et al., 2023; Guo et al., 2023; Wu et al., 2024). Another line of research leverages the success of image-based models (Ho & Salimans, 2022; Nichol et al., 2022; Podell et al., 2023; Avrahami et al., 2022; Brooks et al., 2023) by adapting their image-editing capabilities to ensure temporal consistency during test time (Khachatryan et al., 2023; Geyer et al., 2024; Wu et al., 2024; Qi et al., 2023; Wang et al., 2023). However, inconsistencies accumulate in this frame-by-frame editing process, causing the edited video to deviate significantly from the original source over time. This accumulation of errors makes it challenging to maintain visual coherence and fidelity, especially in long videos. A significant gap remains in addressing both global and local contexts, leading to inaccuracies and inconsistencies across the spatiotemporal dimension.

To address these challenges, we introduce VIA, a unified spatiotemporal video adaptation framework designed for consistent and precise video editing, pushing the boundaries of editing minute-long videos, as shown in Fig. 1. First, our framework introduces a novel *test-time editing adaptation* mechanism that tune the image editing model on dataset generated by itself using the video to be edited, allowing the image editing model to learn associations between specific visual editing directions and corresponding instructions. This significantly enhances semantic comprehension and editing consistency within individual frames. To further improve local consistency, we introduce local latent adaptation to control local edits across frames, ensuring frame consistency before and after editing.

Second, effective editing requires seamless transitions and consistent edits, especially for long videos. To address this, we introduce *spatiotemporal attention adaptation* to maintain global editing coherence across the edited frames. Specifically, we propose *gather-and-swap* to *gather* consistent attention variables from the model's architecture and strategically apply them throughout the video sequence. This approach not only aligns with the continuity of the video but also reinforces the fidelity of the editing process.

Through rigorous evaluation, our methods have demonstrated superior performance compared to existing techniques, delivering significant improvements in both local edit precision and the overall aesthetic quality of the videos. Moreover, our approach is considerably faster than previous methods due to the parallelized swapping process. To the best of our knowledge, we are the first to achieve consistent editing of minute-long videos. Our main contributions are as follows:

- We introduce VIA, a novel framework designed to enable **faithful, consistent, precise, and fast video editing.** Our approach pushes the boundaries of current video editing methods, ensuring both local and global consistency across the entire video.

- We introduce a novel **spatiotemporal attention adaptation** and **test-time adaptation mechanism**, enabling coherent, text-driven video edits by maintaining global consistency across frames and semantic consistency within individual frames, leveraging an image editing model for video editing.
- **Our approach outperforms existing techniques in human evaluation and automatic evaluation**, delivering significantly better performance in terms of editing quality and efficiency.

## 2 Related Work

### 2.1 Text-driven Video Editing

Text-driven video editing is a process of modifying videos according to the user's instructions. Inspired by the remarkable success of text-driven image editing (Avrahami et al., 2022; Brooks et al., 2023; Tumanyan et al., 2023; Sheynin et al., 2023; Zhang et al., 2023), extensive methods have been proposed for video content editing (Ouyang et al., 2024; Feng et al., 2024; Li et al., 2024; Yang et al., 2024; Zhang et al., 2024; Qin et al., 2023; Khachatryan et al., 2023; Geyer et al., 2024; Wu et al., 2024; Qi et al., 2023; Wang et al., 2023; Ku et al., 2024). One paradigm for video editing is to adapt an image-based model to video. For example, Khachatryan et al. (2023) adapts image editing to the video domain without any training or fine-tuning by changing the self-attention mechanisms in Instruct-Pix2Pix to cross-frame attentions. Geyer et al. (2024) explicitly propagates diffusion features based on inter-frame correspondences to enforce consistency in the diffusion feature space. Yang et al. (2023b) construct a neural video field to enable encoding long videos with hundreds of frames in a memory-efficient manner and then update the video field with an image-based model to impart text-driven editing effects. Ku et al. (2024) plug in any existing image editing tools to support an extensive array of video editing tasks. However, these methods are constrained by their ability to maintain global and local consistency, limiting to edit short videos within seconds. To efficiently enable longer video editing, Wu et al. (2024) centers on the concept of anchor-based cross-frame attention, firstly achieving editing 27-second videos. In our work, we built upon this line of work and improve editing consistency, firstly pushing the limits of editing to minutes-long videos.

### 2.2 Spatiotemporal Consistency

Ensuring spatiotemporal consistency is critical for video editing, especially for long videos. Qi et al. (2023) makes the attempt to study and utilize the cross-attention and spatial-temporal self-attention during DDIM inversion. Wang et al. (2023) proposes a spatial regularization module to fidelity to the original video. Park et al. (2024) presents spectral motion alignment (SMA), a framework that learns motion patterns by incorporating frequency-domain regularization, facilitating the learning of whole-frame global motion dynamics, and mitigating spatial artifacts. Ceylan et al. (2023) and Wu et al. (2023) improve the design of spatial attention to cross-frame attention to ensure consistency. In our work, we further ensure consistency inside the anchor-based frames and propose a two-step gather-swap process to adapt spatiotemporal attention for consistent global editing.

## 3 Preliminaries

**Diffusion Models.** In this work, we adapt an image editing model for instruction-based video editing. Given an image $x$, the diffusion process produces a noisy latent $z_t$ from the encoded latent $z = \mathcal{E}(x)$ where the noise level increases over current timestep $t$ over total $T$ steps. A network $\epsilon_\theta$ is trained to minimize the following optimization problem,

$$\min_\theta \mathbb{E}_{y,\epsilon,t}\Big[\big\|\epsilon - \epsilon_\theta(z_t, t, \mathcal{E}(c_I), c_T)\big\|\Big] \tag{1}$$

where $\epsilon \in \mathcal{N}(0,1)$ is the noise added by the diffusion process and $y = (c_T, c_I, x)$ is a triplet of instruction, input image and target image. Here $\epsilon_\theta$ uses a U-Net architecture (Ronneberger et al., 2015), including convolutional blocks, as well as self-attention and cross-attention layers.

**Attention Layer.** The attention layer first computes the attention map using query, $\mathbf{Q} \in \mathbb{R}^{n_q \times d}$, and key, $\mathbf{K} \in \mathbb{R}^{n_k \times d}$ where $d$, $n_q$ and $n_k$ are the hidden dimension and the numbers of the query and key tokens

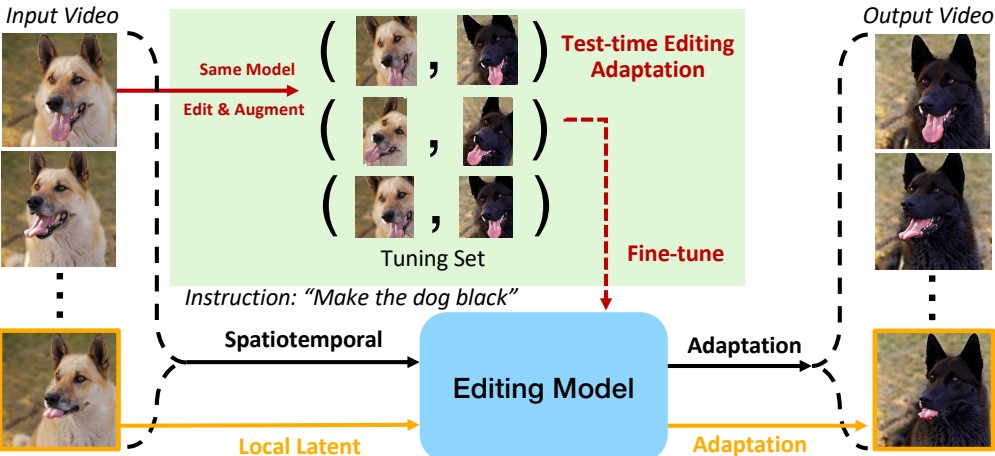

Figure 2: **Overview of Via framework.** For local consistency, Test-time Editing Adaptation finetunes the editing model with augmented editing pairs to ensure consistent editing directions with the text instruction, and Local Latent Adaptation achieves precise editing control and preserves non-target pixels from the input video. For global consistency, Spatiotemporal Adaptation collects and applies key attention variables across all frames.

respectively. Then, the attention map is applied to the value, $\mathbf{V} \in \mathbb{R}^{n \times d}$ as follows:

$$\mathbf{Z}' = \text{Attention}(\mathbf{Q}, \mathbf{K}, \mathbf{V}) = \text{Softmax}(\frac{\mathbf{Q}\mathbf{K}^\top}{\sqrt{d}})\mathbf{V}, \tag{2}$$

$$\mathbf{Q} = \mathbf{Z}\mathbf{W}_q, \quad \mathbf{K} = \mathbf{C}\mathbf{W}_k, \quad \mathbf{V} = \mathbf{C}\mathbf{W}_v, \tag{3}$$

where $\mathbf{W}_q, \mathbf{W}_k, \mathbf{W}_v$ are the projection matrices to map the different inputs to the same hidden dimension $d$. $T$ denotes the matrix transpose. $\mathbf{Z}$ is the hidden state and $\mathbf{C}$ is the condition. For self-attention layers, the condition is the hidden state, while the condition is text conditioning in cross-attention layers.

**Cross-frame Attention.** Given $N$ frames from the source video, cross-frame attention has been employed in video editing by incorporating $\mathbf{K}$ and $\mathbf{V}$ from previous frames into the current frame's editing process (Liu et al., 2023b; Wang et al., 2023; Wu et al., 2024), as shown below:

$$\phi = \text{Softmax}\left(\frac{\mathbf{Q}_{\text{curr}}[\mathbf{K}_{\text{curr}}, \mathbf{K}_{\text{group}}]^{\mathbf{T}}}{\sqrt{d}}\right)[\mathbf{V}_{\text{curr}}, \mathbf{V}_{\text{group}}], \tag{4}$$

where $\mathbf{K}_{\text{group}} = [\mathbf{K}^0, \dots, \mathbf{K}^k]$ and $\mathbf{V}_{\text{group}} = [\mathbf{V}^0, \dots, \mathbf{V}^k]$, and $k$ is the group size. By incorporating $\mathbf{K}_{\text{group}}$ and $\mathbf{V}_{\text{group}}$ during the video editing process for each frame, the temporal consistency is improved. In this paper, we improve cross-frame attention with a two-stage gather-swap process to significantly improve the spatiotemporal consistency.

## 4 The Via Framework

Below, we outline the distinct methodologies that form the foundation of our approach. We introduce a unified framework to tackle key challenges in instruction-guided video editing, with a focus on ensuring editing consistency and spatiotemporal coherence across video frames by leveraging an image editing model, as shown in Fig. 2. For a video to be edited, we first tune the editing direction of the editing model as the test-time adaptation in Sec. 4.1, then edit each frame by Spatiotemporal Adaptation as in Sec. 4.2. With external masks, we could further achieve targeted editing.

### 4.1 Test-Time Editing Adaptation for Local Consistency

When adapting image editing models for video editing, the same instructions must yield consistent semantic interpretations across frames—for example, every frame should exhibit the same degree of darkness when

instructed to *"make it night."* Additionally, non-target elements in each frame must remain unchanged; for instance, a table should remain intact when the instruction is to replace an apple with an orange. To address these challenges, we propose two orthogonal approaches to achieve consistent local editing.

Inspired by DreamBooth (Ruiz et al., 2023), which employs inference-time fine-tuning to associate specific objects with unique textual tokens, we similarly link visual editing outcomes with corresponding instructions, as shown in Fig. 2. We begin with a pipeline to generate the in-domain tuning set without the need for external resources. The image editing model $\Psi$ first edits a randomly sampled frame $S_{\text{root}}$ from the video to be edited to get editing result $E_{\text{root}}$. Then we apply random affine transformations to both the edited frame and source frame. Consider $\mathcal{F}_k$ as affine transformation:

$$T = \{(\mathcal{F}_k(S), \mathcal{F}_k(E), I) \mid \mathcal{F}_k \in \mathcal{F}\} \tag{5}$$

where $\mathcal{F}$ is the set of transformations. The tuning set $T$ consists of triples: source image, edited image, and editing instruction. Then the editing model is tuned on the triplets that is generated by itself from the video to be edited. Therefore, the model learns to map specific visual editing directions to the corresponding instructions for the video. This (i) multiplies training samples, preventing over-fitting to one pose, and (ii) mimics the small viewpoint or scale changes that naturally occur from frame to frame, letting the model learn an edit that stays stable even when the camera or objects shift slightly. The resulting augmented set makes test-time adaptation far more effective at preserving local consistency across the entire video.

For the second challenge, where edits target specific areas, video models often unintentionally affect untargeted regions. In image editing, background preservation involves inverting the source image into latent space and blending it with the generated latent using a mask to control edits (Cao et al., 2023; Gu et al., 2024). However, directly applying this approach to video editing causes severe glitching issues, as the generated areas do not stay aligned across frames. To address this, we propose **Local Latent Adaptation** in the context of video editing. The core behind it is **Progressive Boundary Integration**, which blends the inverted and generated latents at each timestep, confining edits to designated areas while preserving non-targeted regions. Please check Appendix for more details. Our approach ensures strict adherence to editing instructions, focusing solely on specified areas. Our approach smoothly merges source and target latents via linear interpolation between 0 and 1 over the time series. The mathematical representation is given by:

$$\mathbf{M_t}(x, y) = \begin{cases} \mathbf{M}(x, y) \cdot \frac{t}{T}, & \text{if } t \leq T \text{ and } \mathbf{M}(x, y) = 1 \\ \mathbf{M}(x, y), & \text{otherwise} \end{cases} \tag{6}$$

$$\boldsymbol{z}_t^{target} = \mathbf{M_t} \cdot \boldsymbol{z}_t^{edit} + (1 - \mathbf{M_t}) \cdot \boldsymbol{z}_t^{inverted} \tag{7}$$

$$\boldsymbol{z}_{t-1}^{edit} = Sample(\boldsymbol{z}_t^{target}, \Phi, t) \tag{8}$$

Here, $\mathbf{M}$ is the giving binary mask and $\mathbf{M}(x, y)$ is predefined as 1 in a target area and 0 elsewhere. Within this central area, $\mathbf{M}(x, y)$ incrementally decrease from 0 to 1 over $T$ steps, while the values outside this central region remain unchanged. $\Phi$ denotes the attention output within the U-Net architecture at each diffusion step. Note that other parameters such as editing instruction are ignored for simplicity. To assist VIA framework, we built a mask generation process as in the Appendix.

### 4.1.1 Automatic Mask Generation

We present an automated mask generation pipeline aimed at enhancing user experience and streamlining the editing process, particularly for large-scale edits. Editing instructions often specify modifications to specific regions, but current end-to-end models tend to alter unintended areas. To address this, we designed an automated pipeline for mask generation, as illustrated in Fig. 3.

First, a Large Vision-Language Model (GPT-4V in our experiment) is prompted to generate a textual description, $P$, of the region to be modified for each frame. Using this description, we follow Grounded-SAM Ren et al. (2024) to apply the Segment Anything model (Kirillov et al., 2023) and Grounding DINO Liu et al. (2024) to extract a mask that accurately delineates the target area for editing. It is important to note that we did not use GPT-4V during comparisons with baselines in the original paper.

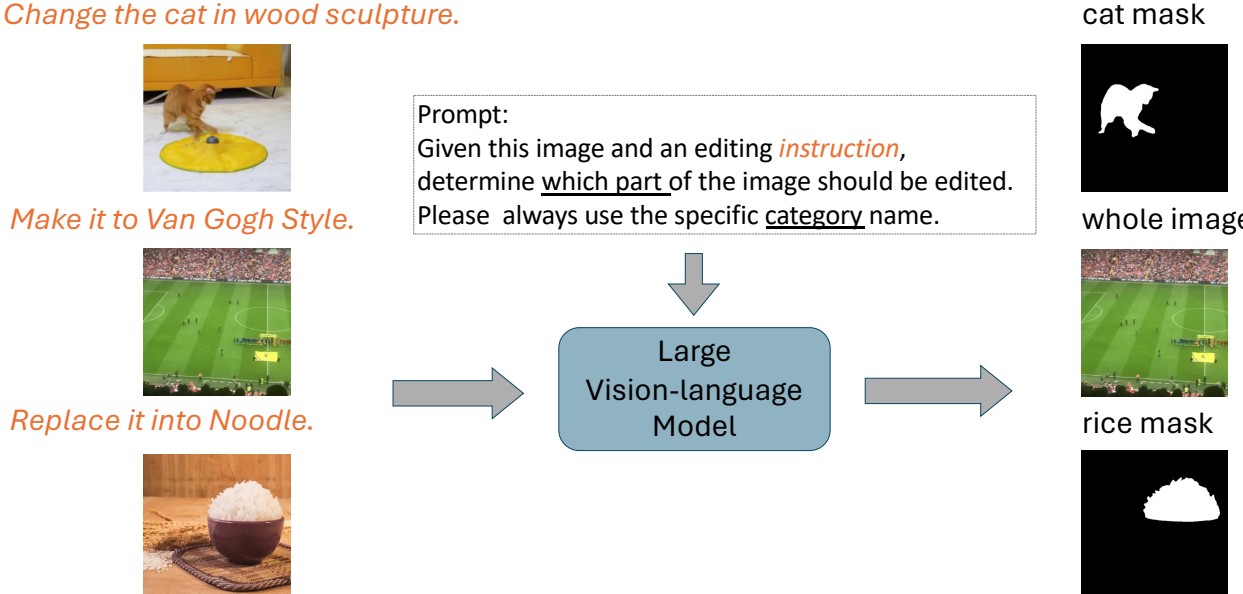

Figure 3: **Automatic mask generation.** A single frame from the video, along with a tailored text prompt encapsulating the editing instruction, is fed into a Large Vision-Language Model (LVLM), such as GPT-4, to generate a text description that specifies the region to be edited. If the designated editing area does not cover the entire image, this text description is then passed into a segmentation model, such as the Segment Anything model, to create a mask for the targeted region. This automated process allows for precise identification of the area to be modified, ensuring that only the relevant portion of the image is edited, while preserving the integrity of the rest of the frame.

In the optimal setting, VIA involves further tuning in the local adaptation process, which some baselines do not utilize. For fairness in comparisons, we degraded our model to use only Spatiotemporal Adaptation during all evaluations. This ensures that our results are directly comparable to baseline models without additional enhancements from local adaptation or the automated mask generation process.

### 4.2 Spatiotemporal Adaptation for Global Consistency

For long video editing, maintaining smooth transitions without glitches or artifacts is essential. Attention variables within the U-net have been found to correlate strongly with the generated content. To ensure consistent global editing, we propose a two-step *gather-and-swap* process to adapt spatiotemporal attention, as illustrated in Fig. 4. In this method, the gathered group is uniformly applied across all frames, ensuring internal coherence throughout the editing process.

Firstly, in the *gather* stage, the model progressively edits the image, with key $\mathbf{K}$ and value $\mathbf{V}$ from previous frames in the group, rather from their own $\mathbf{K}_{\text{curr}}$ and $\mathbf{V}_{\text{curr}}$,

$$\phi = \text{softmax}\left(\frac{\mathbf{Q}_{\text{curr}}\mathbf{K}_{\text{prev}}^T}{\sqrt{d}}\right)\mathbf{V}_{\text{prev}}, \tag{9}$$

$$\mathbf{K}_{\text{group}}^{(t+1)} = [\mathbf{K}_{\text{group}}^{(t)}, \mathbf{K}_{\text{curr}}], \quad \mathbf{V}_{\text{group}}^{(t+1)} = [\mathbf{V}_{\text{group}}^{(t)}, \mathbf{V}_{\text{curr}}] \tag{10}$$

Since $\mathbf{K}_{\text{curr}}$ and $\mathbf{V}_{\text{curr}}$ are calculated by the $\phi$ from the last layer, which already has a stronger dependency on other frames, the saved elements have a stronger consistency with previous group elements, leading to in-group consistency in $\mathbf{K}_{\text{group}}^{(k+1)}$ and $\mathbf{V}_{\text{group}}^{(k+1)}$. Here t denotes the frame index, and group means evenly sampled frames from the source video.

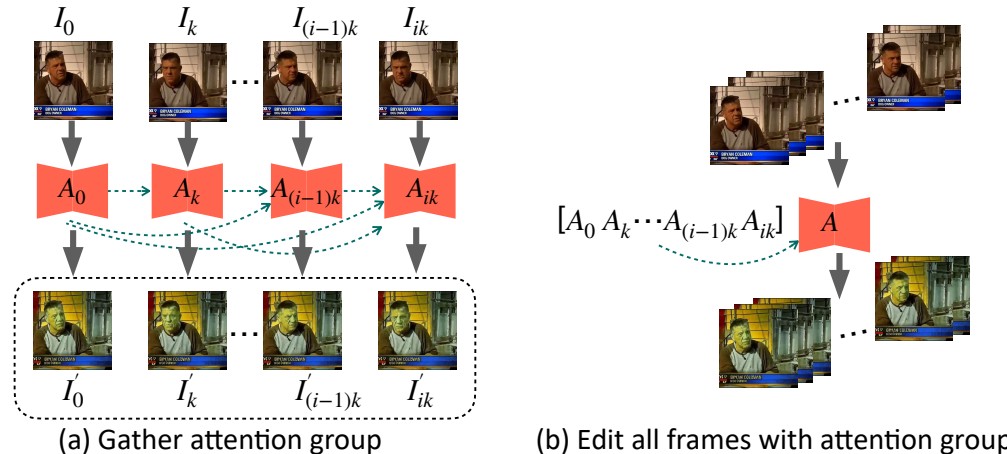

(a) Gather attention group  (b) Edit all frames with attention group

Figure 4: **The *gather-and-swap* process for video editing.** The left part of the diagram illustrates the gathering process. We initially sample $k+1$ frames evenly distributed throughout the video. The first frame undergoes standard editing using an image editing model, during which the attention variables are captured and stored. For each of the subsequent $k$ frames, the attention variable from the preceding frame is swapped in, and its own attention variables are also preserved. In the right part, the collected attention variables from all $k+1$ frames are swapped into the editing process of each frame. This includes applying the previously gathered attention variables to enhance the consistency and quality of edits across the sequence.

In the second stage, we apply the attention group to the editing process of all frames, including those used initially to generate the attention group. Expanding $K$ and $V$ does not change the output, as $QK^T$ remains structured, and multiplication with $V$ keeps the dependency on $Q$ and $V$. Thus, a signal can integrate information from multiple others. This approach resolves the inconsistency in the group frames, where they initially have less dependency on other frames. Throughout the editing process, each frame continues to refrain from using its own attention variables, instead relying on the shared attention group to maintain consistency across the entire video. This ensures that all frames, even the earlier ones, are edited with a global perspective, reducing discrepancies between frames.

$$\phi = \mathrm{softmax}\left(\frac{\mathbf{Q}_{\mathrm{curr}}\mathbf{K}_{\mathrm{group}}^T}{\sqrt{d}}\right)\mathbf{V}_{\mathrm{group}}, \tag{11}$$

In this way, all frames share the same attention group, leading to maximum coherence between the edited frames and enabling the *swap* process to be distributed across multiple GPUs, which significantly reduces editing time. Moreover, while previous work has primarily relied on self-attention for cross-frame consistency, we discovered that cross-attention also plays a crucial role in maintaining coherence. Combining both self-attention and cross-attention mechanisms capturing a broad representation of frame differences and maximizing consistency in the edits. Fig. 4 illustrates the two stages, where **A** represents both **K** and **V**.

Table 1: **Human evaluation results.** We compare our model with five previous open-source methods from three aspects. 'Tie' indicates the two models are on par with each other. Only spatiotemporal adaptation is used when compared with baseline models.

| | Ours | Rerender | Tie | Ours | TokenFlow | Tie | Ours | AnyV2V | Tie | Ours | Video-P2P | Tie | Ours | Tune-A-Video | Tie |
|---|---|---|---|---|---|---|---|---|---|---|---|---|---|---|---|
| Instruction Following | **50.50** | 34.00 | 15.5 | **75.75** | 16.00 | 8.25 | **56.00** | 29.00 | 15.00 | **74.00** | 16.25 | 9.75 | **70.25** | 20.75 | 9.00 |
| Consistency | **47.25** | 35.00 | 17.75 | **38.00** | 31.50 | 30.5 | **53.50** | 23.25 | 23.25 | **80.50** | 9.50 | 10.00 | **68.75** | 20.75 | 10.5 |
| Overall Quality | **53.50** | 29.00 | 17.5 | **61.75** | 22.75 | 15.5 | **63.50** | 30.00 | 6.5 | **63.75** | 22.75 | 13.5 | **56.00** | 22.25 | 21.75 |

## 5   Evaluation

In this paper, we adapt image editing model MGIE (Fu et al., 2024) for video editing. Please refer to the Appendix for performance on other backbone. We conduct both qualitative and human evaluations

Table 2: **Automatic evaluation results.** VIA outperforms open-sourced video editing models in automatic metrics. Only spatiotemporal adaptation is used when compared with baseline models.

|  | VIA | Rerender | TokenFlow | AnyV2V | Video-P2P | Tune-A-Video |
|---|---|---|---|---|---|---|
| Frame-Acc ↑ | **0.869** | 0.734 | 0.587 | 0.533 | 0.587 | 0.601 |
| Tem-Con ↑ | **0.983** | 0.954 | 0.932 | 0.856 | 0.912 | 0.927 |
| Pixel-MSE ↓ | **0.011** | 0.016 | 0.018 | 0.026 | 0.020 | 0.019 |
| Latency(sec) ↓ | **16** | 406 | 450 | 570 | 612 | 529 |

against open-source state-of-the-art baselines, including Fairy (Wu et al., 2024), AnyV2V (Ku et al., 2024), Rerender (Yang et al., 2023a), Tokenflow (Geyer et al., 2024), Video-P2P (Liu et al., 2023b), and Tune-A-Video (Wu et al., 2023). For the comparison with AnyV2V, we use the first edited frame generated by VIA as the starting point for the evaluation. Please refer to the Appendix for details about the implementation process of the baselines. We used 800 videos for the test set, where 400 of them are short video, and the remaining range from 1 minutes to 2 minutes. Short videos are collected from Panda-70M and long videos are from https://www.shutterstock.com/video. We used DDIM for both forward and reverse process.

## 5.1 Quantitative Evaluation

**Human Evaluation.** We began by conducting a human evaluation. Since many baselines are unable to handle long videos, we limited the video length to 4–8 seconds to ensure a fair comparison. All videos were standardized to a frame size of 512x512 pixels. A total of 400 videos were sampled for human evaluation to compare the performance. The evaluation focused on three key criteria: **Instruction Following**, assessing accuracy in executing user commands; **Consistency**, ensuring coherence across frames without abrupt changes; and **Overall Quality**, gauging visual appeal and smoothness. Results in Tab. 4 show that VIA excelled in all metrics compared with other baselines.

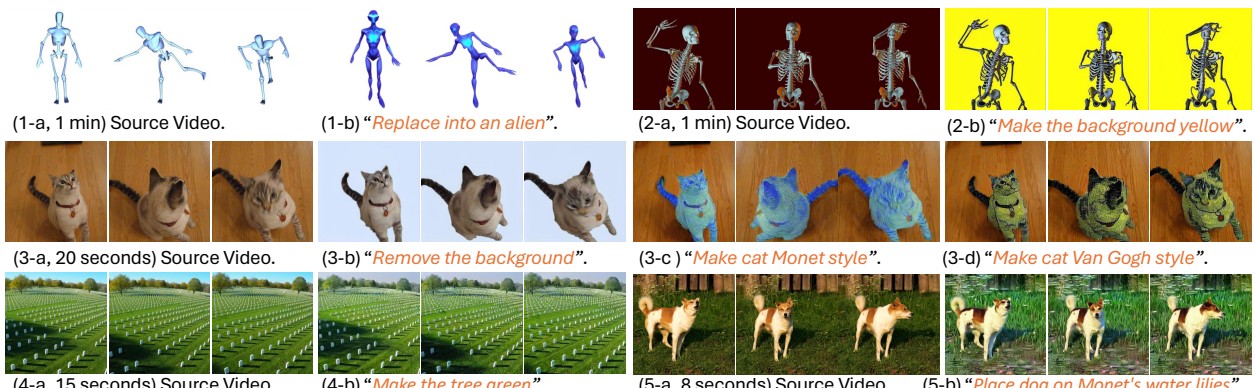

(1-a, 1 min) Source Video.  (1-b) *"Replace into an alien"*.  (2-a, 1 min) Source Video.  (2-b) *"Make the background yellow"*.

(3-a, 20 seconds) Source Video.  (3-b) *"Remove the background"*.  (3-c) *"Make cat Monet style"*.  (3-d) *"Make cat Van Gogh style"*.

(4-a, 15 seconds) Source Video.  (4-b) *"Make the tree green"*.  (5-a, 8 seconds) Source Video.  (5-b) *"Place dog on Monet's water lilies"*.

Figure 5: **Local editing results.** VIA is capable of performing a wide range of localized editing tasks, where only specific regions or pixels within a frame are modified. The video length is introduced in the text below the video frames.

**Automatic Evaluation.** We also conducted automatic evaluation as in Tab. 2. Frame-Acc (Qi et al., 2023; Yang et al., 2023a) measures the percentage of frames where the edited image has a higher CLIP similarity to the target prompt than the source prompt; Tem-Con (Esser et al., 2023) measures the temporal consistency via computing the cosine similarity between all pairs of consecutive frames. Following Ceylan et al. (2023), we also use Pixel-MSE to calculate the difference between the edited frame and its previous frame warped with the optical flow calculated from the source frame pairs. Note that it is normalized by the maximum possible MSE difference. VIA outperformed all other models across these metrics, delivering superior accuracy and consistency while also achieving faster processing speeds. We did not use test-time adaptation for VIA, as some of the baseline models do not inherently benefit from it, which ensured a fair

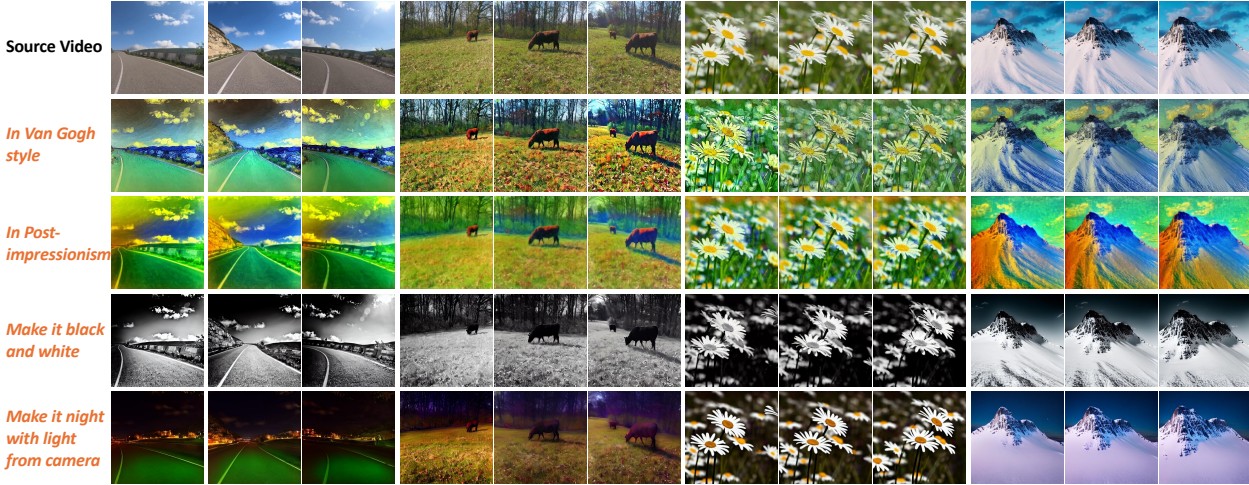

Figure 6: **Global editing results.** VIA demonstrates robust global editing performance across various videos using a consistent set of editing instructions, producing high-quality results. The videos are of length 2-minute, 1-minute video, 30 seconds, and 7 seconds.

comparison. Additionally, we calculated the evaluation latency of the editing process, which was carried out on an A100 machine with 8 GPUs. The global adaptation process could be distributed across multiple GPUs to further accelerate the process. Detailed speed analysis can be found in the Appendix.

## 5.2 Qualitative Results

**Local Editing Results.** Fig. 5 showcases the performance of VIA on various local editing tasks, where only specific parts of the frame are modified. VIA excels at accurately identifying the target area and applying precise edits. VIA demonstrate strong performance on general local editing tasks including both **background modification** and **foreground object modification**. The two 1-min long video in the first row speficially presented its precise control. Besides, VIA enables local stylization, surpassing traditional techniques limited to full-image changes, whose enhanced control opens up new creative possibilities in video editing.

**Global Editing Results.** Fig. 6 highlights the global editing capabilities of VIA across a range of videos. A uniform set of editing instructions was used across different videos, resulting in coherent and visually appealing modifications throughout. The bottom example specifically illustrates VIA's proficiency in understanding and consistently applying visual effects across all frames, ensuring seamless transitions and maintaining the integrity of the visual narrative across the entire video.

**Long Video Editing.** A direct consequence of the high consistency feature in our video editing framework is its proficiency in handling longer videos, as demonstrated throughout this paper. Currently, existing video editing models cannot handle minute-long videos due to architectural limitations, making direct comparisons challenging. To address this, we evaluate long video editing by concatenating individually edited chunks, where VIA significantly outperforms the baselines. For more details, see Sec. 5.3. One of our baselines, Fairy (Wu et al., 2024), has not made their code publicly available, but they report that their model supports videos up to 27 seconds in length. We compare our results on the same video in their website using identical editing instructions, as shown in Fig. 7. VIA demonstrates superior global and local consistency, which can be attributed to our unified adaptation framework.

**Qualitative Comparison.** In Fig. 8, we present two examples of video editing to showcase the performance of VIA in comparison to other models. In the first example, the video depicts rapidly moving clouds against a blue sky, with the instruction to "Set the time to sunset." Despite the swift movement of the clouds, which places a high demand on temporal consistency, VIA demonstrates excellent coherence across frames. The Editing Adaptation process allows VIA to effectively align the visual effect with the concept of

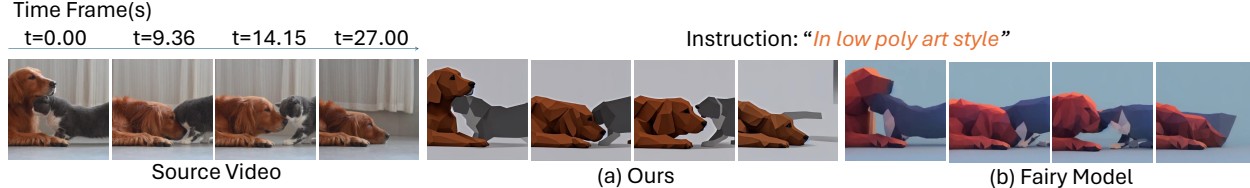

Figure 7: **Comparison with the baseline model on the long video.** We present the editing results from a 27-second video.

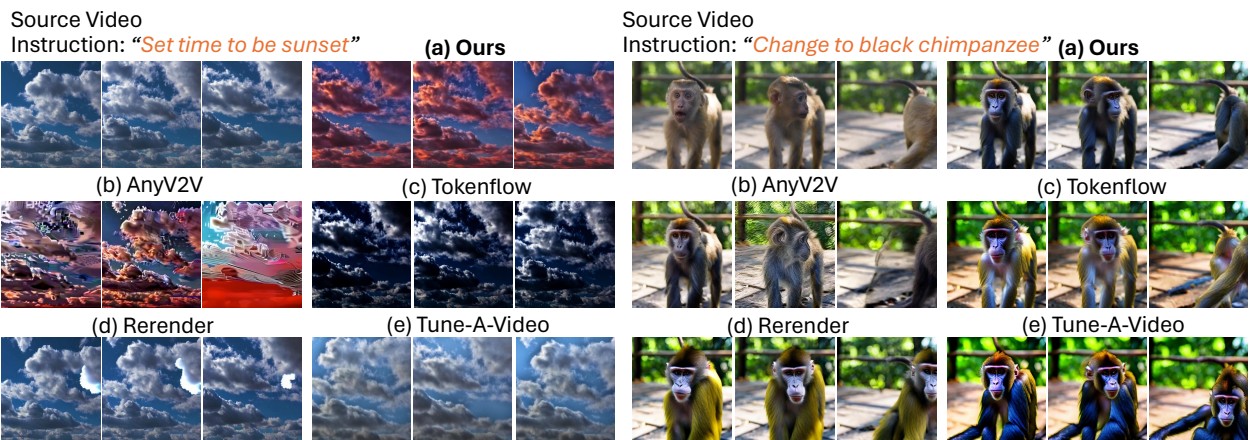

Figure 8: **Qualitative comparison with baselines.** VIA is able to produce consistent editing results.

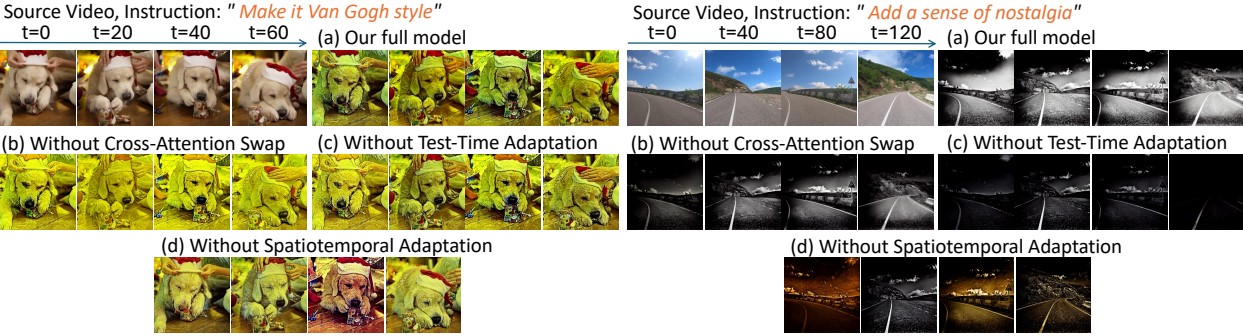

Figure 9: **Ablation Study on components in Via on long video.** In the left example, the hat color and visual style are less consistent without distinct component handling. In contrast, the right example shows a uniform visual style applied consistently across frames, with each component maintaining its appearance. Test-time adaptation ensures stable visual effects that follow the specified instructions. Without the gather-swap technique, object consistency across frames is weakened. Additionally, incorporating cross-attention alongside self-attention improves consistency and reduces artifacts.

"sunset," ensuring smooth and realistic changes. In contrast, other models struggled to execute the command adequately. The AnyV2V model partially achieved the desired visual effect by leveraging the initial frame generated by VIA. On the right, we show an object-swapping example where a monkey moves from within the frame to outside of it. The challenge lies in maintaining a smooth transition from the full subject to a partially visible one. While other methods often introduce artifacts between the edited frames and the original video, VIA seamlessly swaps the subject's identity, preserving visual coherence and continuity throughout the transition.

From this comparison, we found that (1) VIA outperforms the baselines in both editing quality and processing speed. It ensures smooth transitions in edited videos, even when dealing with rapidly moving objects, while

some models, such as AnyV2V, generate noticeable artifacts. (2) VIA demonstrates strong performance in adhering to complex instructions, where other models often struggle. While competing methods experience degraded performance with intricate commands, VIA consistently follows the instructions, applying edits accurately across all frames.

**Ablation on Individual Components.** In Fig. 9, we first present qualitative results for ablation. We analyze the impact of various components of VIA on the editing of long videos. Our experiments indicate that the quality of the initial edited frames plays a critical role in determining the overall visual quality, as information from these root frames propagates throughout the video sequence. Test-time adaptation further enhances the model's ability to closely follow the editing instructions, improving overall consistency. When *gather-and-swap* is omitted and the model relies solely on cross-frame attention, inconsistencies start to emerge between frames. Additionally, although self-attention is commonly employed to ensure consistency, we found that the inclusion of cross-attention significantly improves the quality of video editing. In the left example, the hat color and visual style lack consistency due to the absence of distinct component handling. In contrast, the right example demonstrates a cohesive visual style applied uniformly across frames, with each component retaining its appearance. For additional ablation studies, and analysis on detailed components such as Progressive Boundary Integration, please refer to the Appendix.

Tab. 3 presents a quantitative ablation in which each core component of VIA is removed in turn—Cross-Attention (CA), Test-Time Adaptation (TTA), Spatiotemporal Adaptation (SA), Local Latent Adaptation (LLA), Progressive Boundary Integration (PBI), and the affine augmentations used during TTA—while the full model appears in the leftmost column. The full configuration yields the highest scores across all metrics (e.g., 0.826 for Frame-Acc and 0.942 for Tem-Con on long videos). Eliminating any single module consistently degrades performance: SA and LLA cause the largest drops (up to 0.034 on Frame-Acc and 0.032 on Tem-Con), TTA and PBI each lower accuracy and temporal coherence by roughly 0.02–0.03, and CA or the affine transforms introduce smaller but still noticeable declines. The uniform decline across both long- and short-video benchmarks confirms that every component contributes meaningfully to spatial fidelity and temporal stability, validating the necessity of VIA's full design.

Table 3: **Quantitative Ablation Study.** CA means Cross-Attention; TTA means Test-Time Adaptation; SA means Spatiotemporal Adaptation; LLA means Local Latent Adaptation; PBI means Progressive Boundary Integration. Affline means affline transformation.

|  | VIA | w/o CA | w/o TTA | w/o SA | w/o LLA | w/o PBI | w/o Affine |
|---|---|---|---|---|---|---|---|
| (Long) Frame-Acc ↑ | **0.826** | 0.814 | 0.801 | 0.803 | 0.792 | 0.805 | 0.814 |
| (Long) Tem-Con ↑ | **0.942** | 0.923 | 0.913 | 0.909 | 0.910 | 0.920 | 0.921 |
| (Short) Frame-Acc ↑ | **0.869** | 0.852 | 0.844 | 0.842 | 0.833 | 0.857 | 0.847 |
| (Short) Tem-Con ↑ | **0.983** | 0.952 | 0.943 | 0.928 | 0.955 | 0.968 | 0.966 |

### 5.3 Long Video Comparison

Since prior methods do not support long video editing, we divide long videos into 5-second segments, edit each segment separately, and then concatenate the results. VIA significantly outperforms other baselines by a large margin. However, independently editing each chunk introduces noticeable inconsistencies. As an example shown in Fig. 10, applying AnyV2V Ku et al. (2024) to two consecutive chunks results in visibly different editing effects across segments.

### 5.4 Speed Analysis

VIA not only achieves great performance, but also offers impressive speed. The fine-tuning process takes approximately 1 minute, regardless of the video's length. For the global adaptation process, it takes InstructPix2Pix (Brooks et al., 2023) about 1 second per frame, and MGIE (Fu et al., 2024) around 3 seconds per frame.

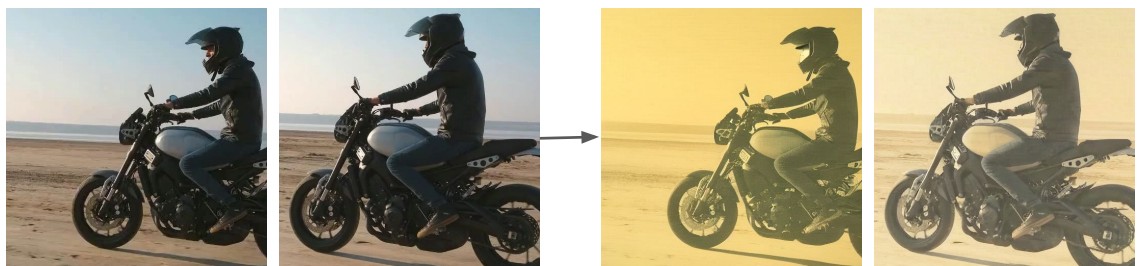

**1st Frame of 1st chunk   1st Frame of 2nd chunk   1st Frame of 1st chunk   1st Frame of 2nd chunk**

Figure 10: Editing results from two consecutive 5-second chunks. The editing instruction is "Change the video to Japanese Woodprint painting." Even with the same model and random seed, the editing results can vary, leading to noticeable inconsistencies in the concatenated video.

Table 4: **Comparison with baselines using concatenated edited videos.** We evaluate our model against five previous open-source methods across three aspects. A 'Tie' indicates comparable performance between models. Since prior methods do not support long video editing, we divide long videos into 5-second segments, edit each segment separately, and then concatenate the results.

| | Ours | Rerender | Tie | Ours | TokenFlow | Tie | Ours | AnyV2V | Tie | Ours | Video-P2P | Tie | Ours | Tune-A-Video | Tie |
|---|---|---|---|---|---|---|---|---|---|---|---|---|---|---|---|
| Instruction Following | **53.50** | 31.00 | 15.50 | **72.75** | 13.00 | 14.25 | **58.00** | 25.00 | 17.00 | **72.50** | 18.50 | 9.00 | **70.25** | 21.25 | 8.50 |
| Consistency | **45.25** | 36.00 | 18.75 | **36.00** | 32.50 | 31.5 | **52.50** | 21.50 | 26.00 | **78.50** | 10.50 | 11.00 | **70.75** | 19.75 | 9.50 |
| Overall Quality | **53.00** | 27.00 | 20.00 | **70.75** | 15.50 | 13.75 | **72.50** | 13.25 | 14.25 | **61.75** | 14.75 | 23.50 | **58.00** | 25.50 | 16.50 |

**Distribution Across GPUs:** Once we gather the frames, the editing for all frames can be performed on different GPUs simultaneously, as the frame editing process only depends on the fixed group frames. We utilize 8 GPUs for processing, which helps manage the load effectively.

**Total Processing Time for a 600-frame video:**

- **MGIE:** 60 (fine-tuning) $+ \frac{3 \times 600}{8} = 285$ seconds.

- **InstructPix2Pix:** 60 (fine-tuning) $+ \frac{1 \times 600}{8} = 135$ seconds.

For the comparison with baselines, where only spatiotemporal adaptation is used (without fine-tuning or local adaptation), the time is:

- **MGIE (without fine-tuning):** $\frac{3 \times 600}{8} = 225$ seconds.

- **InstructPix2Pix (without fine-tuning):** $\frac{1 \times 600}{8} = 75$ seconds.

## 6   Conclusion

This paper introduces a novel video editing framework that tackles the critical challenges of achieving temporal consistency and precise local edits. Our approach surpasses the limitations of traditional frame-by-frame methods, delivering coherent and immersive video experiences. Extensive experiments show that our framework outperforms existing baselines in terms of handling temporal dynamics, ensuring local edit precision, and enhancing overall video aesthetic quality. This advancement paves the way for new possibilities in media production and creative content generation, setting a new benchmark for future developments in video editing technology.

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
