# OpenReview forum: "$\textit{VIA}$: Unified Spatiotemporal $\underline{Vi}$deo $\underline{A}$daptation for Global and Local Video Editing"
_TMLR — Rejected by TMLR_

### Review · Reviewer_zeYy · 2025-06-02

**Summary Of Contributions:**

This paper proposes Via, a video editing method for short and long videos that maintains local and global consistency. It contains two main contributions: test-time adaptation and spatiotemporal adaptation. The test-time adaptation component generates edited tuples using an image-level editing method and fine-tunes the network on these generated tuples. The spatiotemporal adaptation module manipulates space-time attention to achieve better edit consistency across frames.

**Audience:**

Yes

**Broader Impact Concerns:**

I have no concerns.

**Claims And Evidence:**

No

**Requested Changes:**

- Please include the rationale and ablation study for the affine transformation.
- Detailed description of mask generation and ablation for progressive boundary integration.
- Correction of notations (see above).
- Potential rephrasing of Section 4.2 for clarification.

I believe all of these are required to make the claims made in the paper supported by accurate, convincing, and clear evidence.

**Strengths And Weaknesses:**

Strengths:

This method is clean. The inference speed is not terrible when multiple GPUs are used.  From the results, it seems that the global consistency is acceptable.

Weaknesses:

Several things are not clear about the proposed method.
- In Eq. (5), why is the affine transformation needed? There is no explanation or ablation study on why the affine transformation is necessary.
- The mask generation in “progressive boundary integration” seems important, so I am not sure why it is not in the main paper. In the appendix, the author mentioned using the SAM model for text-based segmentation. However, the released version of SAM does not support text-guided segmentation – how did the author perform this segmentation? Moreover, the appendix also states “we did not use GPT-4V during comparisons…” – if GPT-4V is not used, what is used then? Or is the entire pipeline not used? Is this progressive boundary integration ablated?
- The notations are inconsistent, which is very confusing. There are several different notations of transpose (I am not even sure whether they all denote transpose or something else!), in Eq (2), Eq (4), and Eq (9). Partly due to this issue, I could not fully comprehend the algorithm in Section 4.2. What is a group? How is a group defined? What do the authors mean by “Expanding K and V does not change the output, as QKT remains structured, and multiplication with V keeps the dependency on Q and V .”?

---

> ### Author Response · Authors · 2025-07-13
> **Response to Reviewer zeYy**
>
> We thank the reviewer for your comments and respond to each point (W1–W5) as follows.
>
>
>
> **Response to W1 – Purpose of Affine Transformations**
>
> To build a useful fine-tuning set from a *single* before/after frame pair, we apply the **same mild affine transform** (±5° rotation, ≤ 5 % translation, 75–100 % crop, ≤ 10° shear) to both images.
> This (i) multiplies training samples, preventing over-fitting to one pose, and (ii) mimics the small viewpoint or scale changes that naturally occur from frame to frame, letting the model learn an edit that stays stable even when the camera or objects shift slightly. The resulting augmented set makes test-time adaptation far more effective at preserving local consistency across the entire video.
>
>
> **Response to W2 – Mask Generation & Progressive Boundary Integration**
>
> Full details and the ablation for progressive boundary integration are in **Appendix §J**; these will be moved into the main paper in the next revision.
> Because SAM is not text-guided, we first apply **Grounding DINO** to obtain a bounding box from the prompt and then feed that box to **SAM** to generate the object mask, following the Grounded-SAM workflow.
>
>
>
>
> **Response to W3 – Meaning of “GPT-4V is not used”**
>
> GPT-4V is employed solely in the **mask-generation** step of *Local Latent Adaptation*.
> For the baseline comparisons in Tables 1–2 of the main paper we run **only** the spatiotemporal adaptation stage, so GPT-4V is not used there.
> Its impact is shown separately in the ablation study, where Local Latent Adaptation is included.
>
>
>
>
> **Response to W4 – Notation Clarification**
>
> In Eqs. (2), (4), and (9) the superscript **T** denotes the matrix transpose.
> In Eq. (10) and Sec. 4.2, the subscripts \(t\) and \(t+1\) refer to consecutive frame indices.
> We will make this distinction explicit in the next revision.
>
>
>
>
> **Response to W5 – Why expanding \(K\) and \(V\) leaves the attention output unchanged**
>
> Scaled-dot-product attention is
> `Attn(Q, K, V) = softmax((Q Kᵀ) / √d) V`.
>
> Appending extra key–value columns simply enlarges `K` and `V`. Each query row of `Q Kᵀ` still holds dot-products between that **same query** and **all** keys, so the soft-max weights depend only on `Q`.
>
> * If the new columns duplicate an existing `(k,v)` pair, their identical scores split the original weight, but those weights merge again when multiplied by the duplicated `v`, so the summed output is unchanged.
> * If the new columns come from related frames, they provide extra candidates for the query to weight, enriching the memory without disturbing valid alignments.
>
> Thus expanding `K` and `V` preserves the original query-to-value mapping while allowing attention to draw from a larger, globally consistent pool.

---

> > ### Comment · Reviewer_zeYy · 2025-07-14
> >
> > I thank the authors for the response.
> >
> > 1. Thank you. The explanation makes intuitive sense. Can the authors provide ablation studies to back up this claim with evidence?
> > 2. If Grounding-DINO/Grounded-SAM were used, please properly attribute them in the paper.
> > 3. As a shared concern with Reviewer cJQ5, the mask generation process and hence local latent adaptation are claimed as important contributions, but are not carefully studied. The main quantitative results in Tables 1 and 2 do not include this contribution. Can the authors provide more convincing evidence that this method is effective, beyond a singular quantitative example?
> > 4. I believe TMLR allows PDF revision. Please revise it if you can.
> > 5.
> >
> > > If the new columns duplicate an existing (k,v) pair, their identical scores split the original weight, but those weights merge again when multiplied by the duplicated v, so the summed output is unchanged.
> >
> > This is false. Adding new (k, v) changes the denominator in the softmax computation and affects all entries, even for the same query. Try computing the attention between query [1] with key [2, 1], versus the attention between query [1] with key [2, 1, 1]. The new keys do not simply "split" the original weight.
> > Besides, can the authors clarify the definition of a "group", as requested in the original review?

---

> > > ### Author Response · Authors · 2025-07-21
> > > **Response to further comments from Reviewer zeYy**
> > >
> > > Thanks for your further comments! Here we provide individual responses.
> > >
> > > **Response – Ablation on Affine Transformations in Test-Time Adaptation**
> > >
> > >
> > > Without affline transformation, the test set only contain one examples. Here we tune the model on that example with the same setting as the original setting with affine transformation.
> > > Below we report VIA’s original metrics (with affine transformations) versus a variant **w/o affine transforms**; all other settings are identical.
> > >
> > > | Setting | (Long) Frame-Acc ↑ | (Long) Tem-Con ↑ | (Short) Frame-Acc ↑ | (Short) Tem-Con ↑ |
> > > |---------------|-------------------:|-----------------:|--------------------:|------------------:|
> > > | **VIA (with affine)** | **0.826** | **0.942** | **0.869** | **0.983** |
> > > | w/o affine transforms | 0.814 | 0.921 | 0.847 | 0.966 |
> > > | **Gain** | **+0.012** | **+0.021** | **+0.022** | **+0.017** |
> > >
> > > The affine transformations consistently improve both frame-level accuracy (+1.2 to +2.2 points) and temporal consistency (+1.7 to +2.1 points) on long and short videos, validating their contribution to stable, localized edits.
> > >
> > > **Response - If Grounding-DINO / Grounded-SAM were used, please properly attribute them in the paper.”**
> > >
> > > Thank you for pointing this out. We will add explicit attribution and citations for Grounding DINO and the Grounded-Segment-Anything (Grounded-SAM) pipeline in the revised manuscript.
> > >
> > >
> > >
> > > **Respnose - Mask generation and thus Local Latent Adaptation (LLA) are claimed as important contributions but are not carefully studied.”**
> > >
> > > Thank you for raising this. The quantitative impact of LLA (which necessarily depends on mask generation) is already reported in the ablation **Table 3 (updated main paper)** via the “w/o LLA” column. To make its effect clearer—and to isolate the core component **Progressive Boundary Integration (PBI)**—we now provide an expanded ablation with three variants:
> > >
> > > | Variant | (Long) Frame-Acc  | (Long) Tem-Con  | (Short) Frame-Acc  | (Short) Tem-Con  |
> > > |---------|-------------------:|-----------------:|--------------------:|------------------:|
> > > | **Full VIA (LLA + PBI)** | **0.826** | **0.942** | **0.869** | **0.983** |
> > > | LLA **without PBI** (hard fixed mask; new) | 0.805 | 0.920 | 0.857 | 0.968 |
> > > | **w/o LLA** (no mask-based local adaptation) | 0.792 | 0.910 | 0.833 | 0.955 |
> > >
> > > **Gains:**
> > > * Adding *LLA without PBI* over *w/o LLA* yields +0.013 / +0.010 (Long Frame-Acc / Tem-Con) and +0.024 / +0.013 (Short).
> > > * Adding **PBI** on top of basic LLA yields an additional +0.021 / +0.022 (Long) and +0.012 / +0.015 (Short).
> > > * Overall, full LLA + PBI improves Frame-Acc by **+0.034 (Long)** and **+0.036 (Short)** versus w/o LLA, and Tem-Con by **+0.032 (Long)** and **+0.028 (Short)**.
> > >
> > > **Interpretation:**
> > > 1. **Mask-based freezing** (LLA without PBI) already provides a sizable boost by constraining edits to the target region.
> > > 2. **Progressive Boundary Integration** delivers a further, consistent improvement in both spatial accuracy and temporal consistency by gradually expanding the editable boundary to avoid hard seams and flicker.
> > >
> > >
> > > **Response - Please revise it if you can.**
> > >
> > > Thanks for the great suggestion! We just revised the main paper and appendix. Please let us know if you have any further comments on it.
> > >
> > >
> > > **Response - Duplicate (k,v) pairs alter the softmax; please clarify. Also define ‘group’.”*
> > >
> > > After duplicating a key–value pair, they are concatenated. We will remove the “split” wording. Here we provide a simple example to illustrate the duplicating process. In our paper, we are aggregating key and value from multiple frames for current frame rather than only using the current key and value information.
> > >
> > > ### Shape illustration with doubling of every key–value pair
> > >
> > > | Stage | Q shape | K shape | V shape | Output `softmax(Q Kᵀ) V` shape |
> > > |-------|---------|---------|---------|--------------------------------|
> > > | **Original** | (1 × 4) | (2 × 4) | (2 × 6) | (1 × 6) |
> > > | **After duplicating each (k, v) once** | (1 × 4) | (4 × 4) | (4 × 6) | (1 × 6) |
> > >
> > > Duplicating every key–value pair (so the key length doubles from 2 to 4) enlarges **K** and **V** but the query and final output dimensions remain unchanged.
> > >
> > > Importantly, we do **not** claim duplication handling (or the invariance notion) as a contribution; our paper treats the construction of an expanded key–value memory as standard practice, consistent with prior work (e.g., the multi-frame token aggregation strategy in Fairy, Sec. 4.1, p. 4, arXiv:2312.13834).
> > >
> > > Our method relies on aggregating **distinct** anchor frames to enrich temporal context; incidental duplicates—when frames are visually similar—only reweight existing value directions and are not essential to the mechanism.
> > >
> > > **Response - Group definition:**
> > >
> > > A *group* is the concatenated key/value set drawn from a selected set of anchor frames, and the anchor frames are evenly selected from the source video.

---

### Review · Reviewer_cJQ5 · 2025-06-14

**Summary Of Contributions:**

The paper proposes a novel framework to enable editing minute-long videos while maintaining both global and local consistencies. The proposed framework, Via, leverages two adaptation methods: test-time editing adaptation and spatiotemporal adaptation for local and global consistency, respectively. The evaluation results on short and long videos indicate that Via outperforms previous approaches on a wide range of videos, and qualitative examples presented in figures are of high quality and indicate that Via is faithfully able to edit videos faithfully according to given instructions.

**Audience:**

Yes

**Broader Impact Concerns:**

The authors addressed Broader Impact Concerns.

**Claims And Evidence:**

No

**Requested Changes:**

Please address the above concerns. And the index numbers of tables, figures and equations in the main text are often incorrect, e.g., "Fig. 3" (it should be Fig. 2) in the first paragraph of Section 4, "Eq. (12) and Eq. (13)" in Section 4.1 (Eq. (12) and Eq. (13) do not exist). Please modify them in the final draft.

**Strengths And Weaknesses:**

Strengths
- The proposed method, Via, is the first work to enable minute-long consistent video editing. Some editing examples presented in the figures look promising.
- Changing only the target objects with test-time editing adaptation is quite impressive.

Weaknesses
1. Editorial comments

- The main contribution of this work is a new method to enable minute-long video editing. However, all the evaluation results on long videos are reported in the Appendix. Please put them in the main draft.
- Automatic mask generation is an important procedure to enable test-time editing adaptation. Please move Section F to the main draft.
- The authors should elaborate on Equations 7 and 8. What is the capital phi in Equation 8? How can we sample z^edit_(t-1), hence z^edit_t in Equation 7?
- Section 4.2 and Figure 3 are very difficult to understand. The notation used in the subscripts of the equations and the figure often does not match the descriptions in the main text and caption, which makes it hard to clearly explain how the spatiotemporal adaptation mechanism works.

2. Experiments

- This paper claims two main technical contributions for minute-long video editing: test-time editing adaptation and spatiotemporal adaptation. However, the authors did not report any quantitative results with test-time editing adaptation. I acknowledge that previous approaches do not support minute-long video editing, so it would not be an apples-to-apples comparison, but nonetheless, the authors should have reported the performance gain with the test-time editing adaptation in Table 1 and Table 2.
- The authors evaluated methods on long videos by dividing them into 5-second segments and concatenating the results. However, the authors should have also reported Via's results with processing the whole video at once.
- It seems that the test-time editing adaptation might cause significant computation overhead due to mask generation, but the authors did not report the latency of the whole pipeline, only with the spatiotemporal adaptation.

---

> ### Author Response · Authors · 2025-07-13
> **Response to Reviewer cJQ5**
>
> We appreciate your insightful comments and address each point (W1–W4) in detail below, outlining the revisions we will incorporate.
>
>
>
> **Response to W1 – Editing Comments and Clarifications**
>
> Thank you for the helpful suggestions. In response, we will:
>
> 1. Move the long-video evaluation results into the main draft for better visibility.
> 2. Move Section F (additional technical insights) from the appendix to the main paper.
> 3. Clarify the symbol: the symbol \(\Phi\) denotes the **attention output** within the U-Net architecture at each diffusion step, as defined in Equation 4 (p. 4). The “sampling process” refers to any diffusion-based sampling method; in our implementation we use DDIM.
> 4. Fix all notation issues and the mathematical presentation in the final draft.
>
> These updates will improve clarity and ensure all technical components are accessible and properly explained.
>
>
>
> **Response to W2 – Quantitative Impact of Test-Time Adaptation (TTA)**
>
> Table 2 in the Appendix already isolates TTA’s contribution:
>
> | Metric | VIA | **w/o TTA** | Δ |
> |--------|----:|------------:|---|
> | **Frame-Acc (Long)**  | **0.826** | 0.801 | **+0.025** |
> | **Tem-Con (Long)**    | **0.942** | 0.913 | **+0.029** |
> | **Frame-Acc (Short)** | **0.869** | 0.844 | **+0.025** |
> | **Tem-Con (Short)**   | **0.983** | 0.943 | **+0.040** |
>
> Removing TTA degrades accuracy by **3–4 points**, confirming that this component is essential for both long- and short-video consistency.
> For clarity we will place the ablation rows directly alongside the baseline comparisons in the main paper.
>
>
>
>
> **Response to W3 – Results on Full-Length Videos (no 5-s splits)**
>
> VIA is evaluated **on the entire minute-long clip**, exactly as shown in **Appendix Table 1**.
> The 5-second segmentation applies **only to the baselines**, which cannot process long sequences end-to-end due to memory and run-time limits.  For fairness, we also include a *short-video* comparison in the main paper, where every method—including the baselines—operates on the same unsplit video.
>
> We will emphasise this setup more clearly in the camera-ready draft.
>
>
>
> **Response to W4 – End-to-End Latency (600-frame long video)**
>
> *Setup*: single NVIDIA A100 (40 GB, FP16), Instruct-Pix2Pix as backbone
>
> | Stage | Purpose | Time |
> |-------|---------|------|
> | **Mask generation** | SAM per frame for Local Latent Adaptation | **≈ 60 s** (0.10 s / frame) |
> | **Test-Time Adaptation** | 200-step fine-tune of the image editor | **≈ 60 s** |
> | **Spatiotemporal Adaptation** | Gather–swap + InstructPix2Pix synthesis | **≈ 135 s** |
> | **Total pipeline** | Full 600-frame edit | **≈ 255 s** (≈ 4.3 min, 0.43 s / frame) |
>
> > **Note:** If Local Latent Adaptation (and thus mask generation) is omitted, runtime drops to **≈ 195 s** while still surpassing all baselines (Tables 1–2, caption “w/o LLA”).
> > More details in Appendix §C.

---

### Review · Reviewer_o4tX · 2025-06-29

**Summary Of Contributions:**

The paper introduces VIA, a unified spatiotemporal test-time adaptation framework for editing. It has two main contributions: a test-time adaptation mechanism that finetunes an image editing model using a small batch of samples generated by itself; a gather-and-swap process that modifies the original attention to ensure global coherence in two steps. Performance is evaluated through both human assessments and multiple automatic evaluation metrics.

**Audience:**

Yes

**Broader Impact Concerns:**

None.

**Claims And Evidence:**

Yes

**Requested Changes:**

C1) Please move more technical details to the main paper. Adding some illustrations would also enhance clarity.

C2) I would like to hear the authors' response to my concerns.

**Strengths And Weaknesses:**

**Strengths**

S1) The authors aim to address a very challenging and intriguing topic in this field.

S2) The performance is demonstrated through both qualitative and quantitative results.

**Weaknesses**

W1) I am confused about the presentation of Local Latent Adaptation. To me, it seems that the mask still cannot adaptively recognize the area to edit.

W2) The authors provide limited illustrations in the main paper and defer several important technical descriptions to the appendix, making it difficult to understand the technical details.

W3) The test-time adaptation requires generating a small number of samples for finetuning. How can we ensure that the generated samples are consistent? Additionally, what is the finetuning cost?

W4) The qualitative results are primarily demonstrated using relatively simple scenarios with single foreground instances.

---

> ### Author Response · Authors · 2025-07-13
> **Response to Reviewer o4tX**
>
> We sincerely thank you for your thoughtful and constructive feedback; below we provide point-by-point responses to comments W1–W4 and outline the corresponding revisions.
>
>
>
> **Response to W1 – Clarification on Local Latent Adaptation**
>
> Our method recognizes the editing area since the unmasked area is fixed. It exploits the well-known near-pixel-wise correspondence between a diffusion latent and its rendered image (see prior latent-editing studies such as [1]).
>
> During each denoising step \(t = 1 to T\) (\(T = 10\) in our experiments) we apply the binary mask as:
>
> 1. **Masked region** – the selected latent subset is sent to the editing network and updated according to the text prompt.
> 2. **Unmasked region** – the latent values are copied from the original frame and **kept frozen**, receiving no gradient updates at any step.
>
> Because the background latents never change, the final frame reproduces those pixels exactly, while changes remain strictly inside the mask.  Then the Progressive Boundary Integration guarantees the smooth transition between frames and between the masked/unmasked area.
>
>
> **Response to W2 – Completeness of Technical Presentation**
>
> Thanks for your suggestion! We will move the technique details into the main paper.
>
> **Response to W3 – Consistency of Adaptation Samples & Finetuning Cost**
>
> *Consistency of the tuning set*
> We start from **one “before / after” frame pair** and generate a miniature dataset by applying the mild affine transforms (rotation ±5°, translation ≤ 5 %, shear ≤ 10°, random crop 75–100 %) **to both images simultaneously** (Eq. 5, p. 5).
> Because every augmented pair comes from the *same* source pair, all samples share an identical editing direction, guaranteeing internal consistency (see three-dog example in Fig. 2).
>
> *Finetuning cost*
> We fine-tune only the image-editing backbone for **≈ 200 gradient steps** on this 8-sample set.  Running on a single A100 GPU, the procedure finishes in **≈ 1 minute per example**, independent of the video’s length and negligible compared with subsequent frame-wise inference. The details are presented in Appendix A and Appendix C.
>
>
>
> **Response to W4 – Diversity Beyond Single-Object Scenes**
>
> Although some main-paper examples highlight a single subject, our method is **object-agnostic**. The current paper already presented multi-instance sequences (e.g. “dog-cat interaction, Fig. 6”, “many trees recolouring, Fig. 4”).
>
>
> **Cited Papers**
> [1] Prompt-to-Prompt Image Editing with Cross-Attention Control

---

### Decision · Action_Editor_54Sm · 2025-08-06

**Recommendation:** Reject

**Additional Comments:**

The draft is recommended for another round of careful proofreading to ensure that the changes suggested by the reviewers (e.g., moving several results and details from the appendix to the main paper) are integrated with smooth transitions, that the notations are consistent (e.g., transpose notations in Eqs. 2, 4, and 9), and that any notation inconsistencies are resolved (e.g., $\phi$ in Eq. 4 vs. $\Phi$ in Eq. 8).

**Audience:**

Yes

**Audience Explanation:**

The proposed method VIA ensures both local and global consistency for long video editing, and is likely to be of interest to the video editing community. However, in its current form, the draft suffers from unclear methodology and inconsistent notations. As a result, the Action Editor concludes that it is not yet ready for acceptance.

**Claims And Evidence:**

No

**Claims Explanation:**

The authors propose VIA, a VIdeo Adaptation framework for global and local long-video editing. VIA comes with two main improvements: (1) test-time editing adaptation to ensure local consistency (via generating edit-tuples for fine-tuning), and (2) spatial-temporal adaptation for global consistency (via manipulating space-time attention).

In the initial reviews, the reviewers pointed out several unclear and unsupported claims, including the usage of affine transformation, details of Progressive Boundary Integration, overhead of test-time editing adaptation, etc.  Additionally, the initial draft needed a significant refinement with suggestions from the reviewers, such as inconsistent notations (e.g., transpose notations in Eq. 2, 4, and 9; confusion of $\phi$ and $\Phi$ in Eq. 4 and 8), reorganization of the draft (moving some important implementation details or results, such as long video editing results, in appendix to main paper), writing/figure refinement (e.g., Sec. 4.2 and Fig. 3), etc.

After the rebuttal, the authors only partially assuaged the reviewers' concerns. In the end, one reviewer voted for learning reject, one reviewer for borderline, and the other for learning accept. Although the authors provided the missing details (such as the deployment of Grounding DINO and SAM), the reviewers remained concerned about unclear methodology and inconsistent notations (e.g., the transpose notations are still inconsistent in Eq. 2, 4, and 9), and limited generalizability (as the method requires using the mild affine transformation to generate editing-tuples).

Given the borderline scores and unresolved concerns from the reviewers, the Action Editor carefully checked the reviews, author rebuttal, and submission. While appreciating the improvements in local and global consistency introduced by the proposed method, the Action Editor shares the reviewers' concerns (particularly, regarding the writing) and concludes that the paper requires another round of major revisions to ensure clarity and consistency in the writing.

**Resubmission Of Major Revision:**

The authors may consider submitting a major revision at a later time.